# Lipschitz constant estimation of Neural Networks via sparse polynomial optimization

**Fabian Latorre, Paul Rolland and Volkan Cevher**
EPFL, Switzerland
`firstname.lastname@epfl.ch`

## Abstract

We introduce LiPopt, a polynomial optimization framework for computing increasingly tighter upper bounds on the Lipschitz constant of neural networks. The underlying optimization problems boil down to either linear (LP) or semidefinite (SDP) programming. We show how to use the sparse connectivity of a network, to significantly reduce the complexity of computation. This is specially useful for convolutional as well as pruned neural networks. We conduct experiments on networks with random weights as well as networks trained on MNIST, showing that in the particular case of the $\ell_\infty$-Lipschitz constant, our approach yields superior estimates, compared to baselines available in the literature.

## 1 Introduction

We consider a neural network $f_d$ defined by the recursion:

$$f_1(x) := W_1 x \qquad f_i(x) := W_i \sigma(f_{i-1}(x)), \quad i = 2, \ldots, d \tag{1}$$

for an integer $d$ larger than 1, matrices $\{W_i\}_{i=1}^d$ of appropriate dimensions and an *activation function* $\sigma$, understood to be applied element-wise. We refer to $d$ as the *depth*, and we focus on the case where $f_d$ has a single real value as output.

In this work, we address the problem of estimating the *Lipschitz constant* of the network $f_d$. A function $f$ is *Lipschitz continuous* with respect to a norm $\|\cdot\|$ if there exists a constant $L$ such that for all $x, y$ we have $|f(x) - f(y)| \leq L\|x - y\|$. The minimum over all such values satisfying this condition is called the *Lipschitz constant* of $f$ and is denoted by $L(f)$.

The Lipschitz constant of a neural network is of major importance in many successful applications of *deep learning*. In the context of supervised learning, Bartlett et al. (2017) show how it directly correlates with the generalization ability of neural network classifiers, suggesting it as model complexity measure. It also provides a measure of robustness against adversarial perturbations (Szegedy et al., 2014) and can be used to improve such metric (Cisse et al., 2017). Moreover, an upper bound on $L(f_d)$ provides a certificate of robust classification around data points (Weng et al., 2018).

Another example is the discriminator network of the *Wasserstein GAN* (Arjovsky et al., 2017), whose Lipschitz constant is constrained to be at most 1. To handle this constraint, researchers have proposed different methods like heuristic penalties (Gulrajani et al., 2017), upper bounds (Miyato et al., 2018), choice of activation function (Anil et al., 2019), among many others. This line of work has shown that accurate estimation of such constant is key to generating high quality images.

Lower bounds or heuristic estimates of $L(f_d)$ can be used to provide a general sense of how robust a network is, but fail to provide true certificates of robustness to input perturbations. Such certificates require true upper bounds, and are paramount when deploying safety-critical deep reinforcement learning applications (Berkenkamp et al., 2017; Jin & Lavaei, 2018). The trivial upper bound given by the product of layer-wise Lipschitz constants is easy to compute but rather loose and overly pessimistic, providing poor insight into the true robustness of a network (Huster et al., 2018).

Indeed, there is a growing need for methods that provide tighter upper bounds on $L(f_d)$, even at the expense of increased complexity. For example Raghunathan et al. (2018a); Jin & Lavaei (2018); Fazlyab et al. (2019) derive upper bounds based on *semidefinite programming (SDP)*. While expensive

to compute, these type of certificates are in practice surprisingly tight. Our work belongs in this vein of research, and aims to overcome some limitations in the current state-of-the-art.

**Our Contributions.**

▷ We present **LiPopt**, a general approach for upper bounding the Lipschitz constant of a neural network based on a relaxation to a *polynomial optimization problem (POP)* (Lasserre, 2015). This approach requires only that the unit ball be described with polynomial inequalities, which covers the common $\ell_2$- and $\ell_\infty$-norms.

▷ Based on a theorem due to Weisser et al. (2018), we exploit the sparse connectivity of neural network architectures to derive a sequence of linear programs (LPs) of considerably smaller size than their vanilla counterparts. We provide an asymptotic analysis of the size of such programs, in terms of the number of neurons, depth and sparsity of the network.

▷ Focusing on the $\ell_\infty$-norm, we experiment on networks with random weights and networks trained on MNIST (Lecun et al., 1998). We evaluate different configurations of depth, width and sparsity and we show that the proposed sequence of LPs can provide tighter upper bounds on $L(f_d)$ compared to other baselines available in the literature.

**Notation.** We denote by $n_i$ the number of columns of the matrix $W_i$ in the definition (1) of the network. This corresponds to the size of the $i$-th layer, where we identify the input as the first layer. We let $n = n_1 + \ldots + n_d$ be the total number of neurons in the network. For a vector $x$, $\mathrm{Diag}(x)$ denotes the square matrix with $x$ in its diagonal and zeros everywhere else. For an array $X$, $\mathrm{vec}(X)$ is the *flattened* array. The support of a sequence $\mathrm{supp}(\alpha)$ is defined as the set of indices $j$ such that $\alpha_j$ is nonzero. For $x = [x_1, \ldots, x_n]$ and a sequence of nonnegative integers $\gamma = [\gamma_1, \ldots, \gamma_n]$ we denote by $x^\gamma$ the monomial $x_1^{\gamma_1} x_2^{\gamma_2} \ldots x_n^{\gamma_n}$. The set of nonnegative integers is denoted by $\mathbb{N}$.

**Remark.** The definition of network (1) covers typical architectures composed of dense and convolutional layers. In general, our proposed approach can be readily extended with minor modifications to any directed acyclic computation graph e.g., residual network architectures (He et al., 2016).

## 2 POLYNOMIAL OPTIMIZATION FORMULATION

In this section we derive an upper bound on $L(f_d)$ given by the value of a POP, i.e. the minimum value of a polynomial subject to polynomial inequalities. Our starting point is the following theorem, which casts $L(f)$ as an optimization problem:

**Theorem 1.** *Let $f$ be a differentiable and Lipschitz continuous function on an open, convex subset $\mathcal{X}$ of an euclidean space. Let $\|\cdot\|_*$ be the dual norm. The Lipschitz constant of $f$ is given by*

$$L(f) = \sup_{x \in \mathcal{X}} \|\nabla f(x)\|_* \tag{2}$$

For completeness, we provide a proof in appendix A. In our setting, we assume that the activation function $\sigma$ is Lipschitz continuous and differentiable. In this case, the assumptions of Theorem 1 are fulfilled because $f_d$ is a composition of activations and linear transformations. The differentiability assumption rules out the common ReLU activation $\sigma(x) = \max\{0, x\}$, but allows many others such as the exponential linear unit (ELU) (Clevert et al., 2015) or the softplus.

Using the chain rule, the compositional structure of $f_d$ yields the following formula for its gradient:

$$\nabla f_d(x) = W_1^T \prod_{i=1}^{d-1} \mathrm{Diag}(\sigma'(f_i(x))) W_{i+1}^T \tag{3}$$

For every $i = 1, \ldots, d-1$ we introduce a variable $s_i = \sigma'(f_i(x))$ corresponding to the derivative of $\sigma$ at the $i$-th hidden layer of the network. For activation functions like ELU or softplus, their derivative is bounded between 0 and 1, which implies that $0 \leq s_i \leq 1$. This bound together with the definition of the dual norm $\|x\|_* := \sup_{\|t\| \leq 1} t^T x$ implies the following upper bound of $L(f_d)$:

$$L(f_d) \leq \max \left\{ t^T W_1^T \prod_{i=1}^{d-1} \mathrm{Diag}(s_i) W_{i+1}^T : 0 \leq s_i \leq 1, \|t\| \leq 1 \right\} \tag{4}$$

We will refer to the polynomial objective of this problem as the ***norm-gradient polynomial*** of the network $f_d$, a central object of study in this work.

For some frequently used $\ell_p$-norms, the constraint $\|t\|_p \leq 1$ can be written with polynomial inequalities. In the rest of this work, **we use exclusively the $\ell_\infty$-norm** for which $\|t\|_\infty \leq 1$ is equivalent to the polynomial inequalities $-1 \leq t_i \leq 1$, for $i = 1, \ldots, n_1$. However, note that when $p \geq 2$ is a positive even integer, $\|t\|_p \leq 1$ is equivalent to a single polynomial inequality $\|t\|_p^p \leq 1$, and our proposed approach can be adapted with minimal modifications.

In such cases, the optimization problem in the right-hand side of (4) is a POP. Optimization of polynomials is a NP-hard problem and we do not expect to have efficient algorithms for solving (4) in this general form. In the next sections we describe **LiPopt**: a systematic way of obtaining an upper bound on $L(f_d)$ via tractable approximation methods of the POP (4).

**Local estimation.** In many practical escenarios, we have additional bounds on the input of the network. For example, in the case of image classification tasks, valid input is constrained in a hypercube. In the robustness certification task, we are interested in all possible input in a $\epsilon$-ball around some data point. In those cases, it is interesting to compute a *local Lipschitz constant*, that is, the Lipschitz constant of a function restricted to a subset.

We can achieve this by deriving tighter bounds $0 \leq l_i \leq s_i \leq u_i \leq 1$, as a consequence of the restricted input (see for example, Algorithm 1 in Wong & Kolter (2018)). By incorporating this knowledge in the optimization problem (4) we obtain bounds on local Lipschitz constants of $f_d$. We study this setting and provide numerical experiments in section 7.3.

**Choice of norm.** We highlight the importance of computing good upper bounds on $L(f_d)$ with respect to the $\ell_\infty$-norm. It is one of the most commonly used norms to assess robustness in the adversarial examples literature. Moreover, it has been shown that, in practice, $\ell_\infty$-norm robust networks are also robust in other more plausible measures of perceptibility, like the Wasserstein distance (Wong et al., 2019). This motivates our focus on this choice.

## 3 HIERARCHICAL SOLUTION BASED ON A POLYNOMIAL POSITIVITY CERTIFICATE

For ease of exposition, we rewrite (4) as a POP constrained in $[0, 1]^n$ using the substitution $s_0 := (t+1)/2$. Denote by $p$ the norm-gradient polynomial, and let $x = [s_0, \ldots, s_{d-1}]$ be the concatenation of all variables. Polynomial optimization methods (Lasserre, 2015) start from the observation that a value $\lambda$ is an upper bound for $p$ over a set $K$ if and only if the polynomial $\lambda - p$ is positive over $K$.

In **LiPopt**, we will employ a well-known classical result in algebraic geometry, the so-called *Krivine's positivity certificate*[1], but in theory we can use any positivity certificate like sum-of-squares (SOS). The following is a straightforward adaptation of Krivine's certificate to our setting:

**Theorem 2.** *(Adapted from Krivine (1964); Stengle (1974); Handelman (1988)) If the polynomial $\lambda - p$ is strictly positive on $[0, 1]^n$, then there exist finitely many positive weights $c_{\alpha\beta}$ such that*

$$\lambda - p = \sum_{(\alpha,\beta) \in \mathbb{N}^{2n}} c_{\alpha\beta} h_{\alpha\beta}, \qquad h_{\alpha\beta}(x) := \prod_{j=1}^{n} x_j^{\alpha_j} (1 - x_j)^{\beta_j} \tag{5}$$

By truncating the degree of Krivine's positivity certificate (Theorem 2) and minimizing over all possible upper bounds $\lambda$ we obtain a *hierarchy* of LP problems (Lasserre, 2015, Section 9):

$$\theta_k := \min_{c \geq 0, \lambda} \left\{ \lambda : \lambda - p = \sum_{(\alpha,\beta) \in \mathbb{N}_k^{2n}} c_{\alpha\beta} h_{\alpha\beta} \right\} \tag{6}$$

where $\mathbb{N}_k^{2n}$ is the set of nonnegative integer sequences of length $2n$ adding up to at most $k$. This is indeed a sequence of LPs as the polynomial equality constraint can be implemented by equating coefficients in the canonical monomial basis. For this polynomial equality to be feasible, the degree

---

[1]also known as *Krivine's Positivstellensatz*

of the certificate has to be at least that of the norm-gradient polynomial $p$, which is equal to the depth $d$. This implies that the first nontrivial bound ($\theta_k < \infty$) corresponds to $k = d$.

The sequence $\{\theta_k\}_{k=1}^{\infty}$ is non-increasing and converges to the maximum of the upper bound (4). Note that for any level of the hierarchy, the solution of the LP (6) provides a valid upper bound on $L(f_d)$.

An advantage of using Krivine's positivity certificate over SOS is that one obtains an LP hierarchy (rather than SDP), for which commercial solvers can reliably handle a large instances. Other positivity certificates offering a similar advantage are the DSOS and SDSOS hierarchies (Ahmadi & Majumdar, 2019), which boil down to LP or *second order cone programming* (SOCP), respectively.

**Drawback.** The size of the LPs given by Krivine's positivity certificate can become quite large. The dimension of the variable $c$ is $|\mathbb{N}_k^{2n}| = \mathcal{O}(n^k)$. For reference, if we consider the MNIST dataset and a one-hidden-layer network with 100 neurons we have $|\mathbb{N}_2^{2n}| \approx 1.5 \times 10^6$ while $|\mathbb{N}_3^{2n}| \approx 9.3 \times 10^8$. To make this approach more scalable, in the next section we exploit the sparsity of the polynomial $p$ to find LPs of drastically smaller size than (6), but with similar approximation properties.

**Remark.** In order to compute upper bounds for local Lipschitz constants, first obtain tighter bounds $0 \le l_i \le s_i \le u_i$ and then perform the change of variables $\widetilde{s}_i = (s_i - l_i)/(u_i - l_i)$ to rewrite the problem (4) as a POP constrained on $[0,1]^n$.

## 4 REDUCING THE NUMBER OF VARIABLES

Many neural network architectures, like those composed of convolutional layers, have a highly sparse connectivity between neurons. Moreover, it has been empirically observed that up to 90% of network weights can be *pruned* (set to zero) without harming accuracy (Frankle & Carbin, 2019). In such cases their norm-gradient polynomial has a special structure that allows polynomial positivity certificates of smaller size than the one given by Krivine's positivity certificate (Theorem 2).

In this section, we describe an implementation of **LiPopt** (Algorithm 1) that exploits the sparsity of the network to decrease the complexity of the LPs (6) given by the Krivine's positivity certificate. In this way, we obtain upper bounds on $L(f_d)$ that require less computation and memory. Let us start with the definition of a *valid sparsity pattern*:

**Definition 1.** *Let $I = \{1, \ldots, n\}$ and $p$ be a polynomial with variable $x \in \mathbb{R}^n$. A valid sparsity pattern of $p$ is a sequence $\{I_i\}_{i=1}^m$ of subsets of $I$, called cliques, such that $\bigcup_{i=1}^m I_i = I$ and:*

> ▷ *$p = \sum_{i=1}^m p_i$ where $p_i$ is a polynomial that depends only on the variables $\{x_j : j \in I_i\}$*

> ▷ *for all $i = 1, \ldots, m-1$ there is an $l \le i$ such that $(I_{i+1} \cap \bigcup_{r=1}^i I_r) \subseteq I_l$*

When the polynomial objective $p$ in a POP has a valid sparsity pattern, there is an extension of Theorem 2 due to Weisser et al. (2018), providing a smaller positivity certificate for $\lambda - p$ over $[0,1]^n$. We refer to it as the *sparse Krivine's certificate* and we include it here for completeness:

**Theorem 3** (Adapted from Weisser et al. (2018)). *Let a polynomial $p$ have a valid sparsity pattern $\{I_i\}_{i=1}^m$. Define $N_i$ as the set of sequences $(\alpha, \beta) \in \mathbb{N}^{2n}$ where the support of both $\alpha$ and $\beta$ is contained in $I_i$. If $\lambda - p$ is strictly positive over $K = [0,1]^n$, there exist finitely many positive weights $c_{\alpha\beta}$ such that*

$$\lambda - p = \sum_{i=1}^m h_i, \qquad h_i = \sum_{(\alpha,\beta) \in N_i} c_{\alpha\beta} h_{\alpha\beta} \qquad (7)$$

*where the polynomials $h_{\alpha\beta}$ are defined as in (5).*

The sparse Krivine's certificate can be used like the general version (Theorem 2) to derive a sequence of LPs approximating the upper bound on $L(f_d)$ stated in (4). However, the number of different polynomials $h_{\alpha\beta}$ of degree at most $k$ appearing in the sparse certificate can be drastically smaller, the amount of which determines how *good* the sparsity pattern is.

We introduce a graph that depends on the network $f_d$, from which we will extract a sparsity pattern for the norm-gradient polynomial of a network.

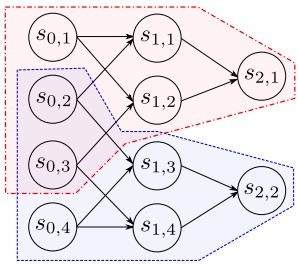

Figure 1: Sparsity pattern of Proposition 1 for a network of depth three.

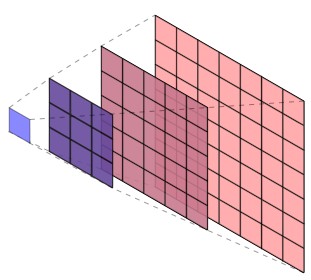

Figure 2: Structure of one set in the sparsity pattern from Proposition 1 for a network with 2D convolutional layers with $3 \times 3$ filters.

**Definition 2.** *Let $f_d$ be a network with weights $\{W_i\}_{i=1}^d$. Define a directed graph $G_d = (V, E)$ as:*

$$V = \{s_{i,j} : 0 \le i \le d-1, \, 1 \le j \le n_i\}$$
$$E = \{(s_{i,j}, s_{i+1,k}) : 0 \le i \le d-2, [W_i]_{k,j} \ne 0\}$$

(8)

*which we call the computational graph of the network $f_d$.*

In the graph $G_d$ the vertex $s_{(i,j)}$ represents the $j$-th neuron in the $i$-th layer. There is a directed edge between two neurons in consecutive layers if they are joined by a nonzero weight in the network. The following result shows that for fully connected networks we can extract a valid sparsity pattern from this graph. We relegate the proof to appendix B.

**Proposition 1.** *Let $f_d$ be a dense network (all weights are nonzero). The following sets, indexed by $i = 1, \dots, n_d$, form a valid sparsity pattern for the norm-gradient polynomial of the network $f_d$:*

$$I_i := \{s_{(d-1,i)}\} \cup \{s_{(j,k)} : \text{ there exists a directed path from } s_{(j,k)} \text{ to } s_{(d-1,i)} \text{ in } G_d\}$$

(9)

We refer to this as the *sparsity pattern induced by $G_d$*. An example is depicted in in Figure 1.

**Remark.** When the network is not dense, the the second condition (Definition 1) for the sparsity pattern (9) to be valid might not hold. In that case we lose the guarantee that the values of the corresponding LPs converge to the maximum of the POP (4). Nevertheless, it still provides a valid positivity certificate that we use to upper bound $L(f_d)$. In Section 7 we show that in practice it provides upper bounds of good enough quality. If needed, a valid sparsity pattern can be obtained via a chordal completion of the *correlative sparsity graph* of the POP (Waki et al., 2006).

We now quantify how good this sparsity pattern is. Let $s$ be the size of the largest clique in a sparsity pattern, and let $N_{i,k}$ be the subset of $N_i$ (defined in Theorem 3) composed of sequences summing up to $k$. The number of different polynomials for the $k$-th LP in the hierarchy given by the sparse Krivine's certificate can be bounded as follows:

$$\left| \bigcup_{i=1}^m N_{i,k} \right| \le \sum_{i=1}^m \binom{2|I_i| + k}{k} = \mathcal{O}\left(m s^k\right)$$

(10)

We immediately see that the dependence on the number of cliques $m$ is really mild (linear) but the size of the cliques as well as the degree of the hierarchy can really impact the size of the optimization problem. Nevertheless, this upper bound can be quite loose; polynomials $h_{\alpha\beta}$ that depend only on variables in the intersection of two or more cliques are counted more than once.

The number of cliques given in the sparsity pattern induced by $G_d$ is equal to the size of the last layer $m = n_d$ and the size of each clique depends on the particular implementation of the network. We now study different architectures that could arise in practice, and determine the amount of polynomials in their sparse Krivine's certificate.

**Fully connected networks.** Even in the case of a network with all nonzero connections, the sparsity pattern induced by $G_d$ decreases the size of the LPs when compared to Krivine's certificate. In this case the cliques have size $n_1 + \dots + n_{d-1} + 1$ but they all have the same common intersection equal to all neurons up to the second-to-last hidden layer. A straightforward counting argument shows that the total number of polynomials is $\mathcal{O}(n(n_1 + \dots + n_{d-1} + 1)^{k-1})$, improving the upper bound (10).

**Unstructured sparsity.** In the case of networks obtained by pruning (Hanson & Pratt, 1989) or generated randomly from a distribution over graphs (Xie et al., 2019), the sparsity pattern can be arbitrary. In this case the size of the resulting LPs varies at runtime. Under the layer-wise assumption that any neuron is connected to at most $r$ neurons in the previous layer, the size of the cliques in (9) is bounded as $s = \mathcal{O}(r^d)$. This estimate has an exponential dependency on the depth but ignores that many neurons might share connections to the same inputs in the previous layer, thus being potentially loose. The bound (10) implies that the number of different polynomials is $\mathcal{O}(n_d r^{dk})$.

**2D Convolutional networks.** The sparsity in the weight matrices of convolutional layers has a certain *local structure*; neurons are connected to contiguous inputs in the previous layer. Adjacent neurons also have many input pixels in common (see Figure 2). Assuming a constant number of channels per layer, the size of the cliques in (9) is $\mathcal{O}(d^3)$. Intuitively, such number is proportional to the volume of the pyramid depicted in Figure 2 where each dimension depends linearly on $d$. Using (10) we get that there are $\mathcal{O}(n_d d^{3k})$ different polynomials in the sparse Krivine's certificate. This is a drastic decrease in complexity when compared to the unstructured sparsity case.

The use of sparsity in polynomial optimization preceeds Theorem 3 (Weisser et al., 2018). First studied in the context of sum-of-squares by Kojima et al. (2005) and further refined in Waki et al. (2006); Lasserre (2006) (and references therein), it has found applications in safety verification (Yang et al., 2016; Zhang et al., 2018), sensor localization Wang et al. (2006), optimal power flow (Ghaddar et al., 2015) and many others. Our work fits precisely into this set of important applications.

---

**Algorithm 1 LiPopt** for ELU activations and sparsity pattern

**Input:** matrices $\{W_i\}_{i=1}^d$, sparsity pattern $\{I_i\}_{i=1}^m$, hierarchy degree $k$.

1: $p \leftarrow (2s_0 - 1)^T W_1^T \prod_{i=1}^{d-1} \mathrm{Diag}(s_i) W_{i+1}^T$        $\triangleright$ compute norm-gradient polynomial
2: $x \leftarrow [s_0, \ldots, s_{d-1}]$
3: $b \leftarrow [b_\gamma : \gamma \in \mathbb{N}_k^n]$ where $p(x) = \sum_{\gamma \in \mathbb{N}_k^n} b_\gamma x^\gamma$        $\triangleright$ compute coefficients of $p$ in basis
4: **for** $i = 1, \ldots, m$ **do**
5:      $N_{i,k} \leftarrow \{(\alpha, \beta) \in \mathbb{N}_k^{2n} : \mathrm{supp}(\alpha) \cap \mathrm{supp}(\beta) \subseteq I_i\}$
6: $\widetilde{N}_k \leftarrow \cup_{i=1}^m N_{i,k}$
7: $h \leftarrow \sum_{(\alpha,\beta) \in \widetilde{N}} c_{\alpha\beta} h_{\alpha\beta}$        $\triangleright$ compute positivity certificate
8: $c \leftarrow [c_{\alpha\beta} : (\alpha, \beta) \in \widetilde{N}_k]$;    $y \leftarrow [\lambda, c]$        $\triangleright$ linear program variables
9: $Z \leftarrow [z_\gamma]_{\gamma \in \mathbb{N}_k^n}$ where $\lambda - h(x) = \sum_{\gamma \in \mathbb{N}_k^n} (z_\gamma^T y) x^\gamma$    $\triangleright$ compute coefficients of $\lambda - h$ in basis
    **return** $\min\{\lambda : b = Zy, \ y = [\lambda, c], \ c \geq 0\}$        $\triangleright$ solve LP

---

## 5   QCQP REFORMULATION AND SHOR'S SDP RELAXATION

Another way of upper bounding $L(f_d)$ comes from a further relaxation of (4) to an SDP. We consider the following equivalent formulation where the variables $s_i$ are normalized to lie in the interval $[-1, 1]$, and we rename $t = s_0$:

$$L(f_d) \leq \max \left\{ \frac{1}{2^{d-1}} s_0^T W_1^T \prod_{i=1}^{d-1} \mathrm{Diag}(s_i + 1) W_{i+1}^T : -1 \leq s_i \leq 1 \right\} \tag{11}$$

Any polynomial optimization problem like (11) can be cast as a (possibly non-convex) *quadratically constrained quadratic program* (QCQP) by introducing new variables and quadratic constraints. This is a well-known procedure described in Park & Boyd (2017, Section 2.1). When $d = 2$ problem (11) is already a QCQP (for the $\ell_\infty$ and $\ell_2$-norm cases) and no modification is necessary.

**QCQP reformulation.** We illustrate the case $d = 3$ where we have the variables $s_1, s_2$ corresponding to the first and second hidden layer and a variable $s_0$ corresponding to the input. The norm-gradient polynomial in this case is cubic, and it can be rewritten as a quadratic polynomial by introducing new variables corresponding to the product of the first and second hidden layer variables.

More precisely the introduction of a variable $s_{1,2}$ with quadratic constraint $s_{1,2} = \mathrm{vec}(s_1 s_2^T)$ allows us to write the objective (11) as a quadratic polynomial. The problem then becomes a QCQP with variable $y = [1, s_0, s_1, s_2, s_{1,2}]$ of dimension $1 + n + n_1 n_2$.

**SDP relaxation.** Any quadratic objective and constraints can then be relaxed to linear constraints on the positive semidefinite variable $yy^T = X \succcurlyeq 0$ yielding the so-called *Shor's relaxation* of (11) (Park & Boyd, 2017, Section 3.3). When $d = 2$ the resulting SDP corresponds precisely to the one studied in Raghunathan et al. (2018a). This resolves a common misconception (Raghunathan et al., 2018b) that this approach is only limited to networks with one hidden layer.

Note that in our setting we are only interested in the optimal value rather than the optimizers, so there is no need to extract a solution for (11) from that of the SDP relaxation.

**Drawback.** This approach includes a further relaxation step from (11), thus being fundamentally limited in how tightly it can upper bound the value of $L(f_d)$. Moreover when compared to LP solvers, off-the-shelf semidefinite programming solvers are, in general, much more limited in the number of variables they can efficiently handle.

In the case $d = 2$ this relaxation provides a constant factor approximation to the original QCQP (Ye, 1999). Further approximation quality results for such hierarchical optimization approaches to NP-hard problems are out of the scope of this work.

**Relation to sum-of-squares.** The QCQP approach might appear fundamentaly different to the hierarchical optimization approaches to POPs, like the one described in Section 3. However, it is known that Shor's SDP relaxation corresponds exactly to the first degree of the SOS hierarchical SDP solution to the QCQP relaxation (Lasserre, 2000). Thus, the approach in section 3 and the one in this section are, in essence, the same; they only differ in the choice of polynomial positivity certificate.

## 6  RELATED WORK

Estimation of $L(f_d)$ with $\ell_2$-norm is studied by Virmaux & Scaman (2018); Combettes & Pesquet (2019); Fazlyab et al. (2019); Jin & Lavaei (2018). The method **SeqLip** proposed in Virmaux & Scaman (2018) has the drawback of not providing true upper bounds. It is in fact a heuristic method for solving (4) but which provides no guarantees and thus can not be used for robustness certification. In contrast the **LipSDP** method of Fazlyab et al. (2019) provides true upper bounds on $L(f_d)$ and in practice shows superior performance over both **SeqLip** and **CPLip** (Combettes & Pesquet, 2019).

Despite the accurate estimation of **LipSDP**, its formulation is limited to the $\ell_2$-norm. The only estimate available for other $\ell_p$-norms comes from the equivalence of norms in euclidean spaces. For instance, we can obtain an upper bound for the $\ell_\infty$-norm after multiplying the $\ell_2$ Lipschitz constant upper bound by the square root of the input dimension. The resulting bound can be rather loose and our experiments in section 7 confirm the issue. In contrast, our proposed approach **LiPopt** can acommodate any norm whose unit ball can be described via polynomial inequalities.

Let us point to one key advantage of **LiPopt**, compared to **LipSDP** (Jin & Lavaei, 2018; Fazlyab et al., 2019). In the context of robustness certification we are given a sample $x^\natural$ and a ball of radius $\epsilon$ around it. Computing an upper bound on the local Lipschitz constant in this subset, rather than a global one, can provide a larger region of certified robustness. Taking into account the restricted domain we can refine the bounds in our POP (see remark in section 1). This potentially yields a tighter estimate of the local Lipschitz constant. On the other hand, it is not clear how to include such additional information in **LipSDP**, which only computes one global bound on the Lipschitz constant for the unconstrained network.

Raghunathan et al. (2018a) find an upper bound for $L(f_d)$ with $\ell_\infty$ metric starting from problem (4) but only in the context of one-hidden-layer networks ($d = 2$). To compute such bound they use its corresponding Shor's relaxation and obtain as a byproduct a differentiable regularizer for training networks. They claim such approach is limited to the setting $d = 2$ but, as we remark in section 5, it is just a particular instance of the SDP relaxation method for QCQPs arising from a polynomial optimization problem. We find that this method fits into the **LiPopt** framework, using SOS certificates instead of Krivine's. We expect that the SDP-based bounds described in 5 can also be used as regularizers promoting robustness.

Weng et al. (2018) provide an upper bound on the local Lipschitz constant for networks based on a sequence of ad-hoc bounding arguments, which are particular to the choice of ReLU activation function. In contrast, our approach applies in general to activations whose derivative is bounded.

# 7 EXPERIMENTS

We consider the following estimators of $L(f_d)$ with respect to the $\ell_\infty$ norm:

| Name | Description |
|---|---|
| **SDP** | Upper bound arising from the solution of the SDP relaxation described in Section 5 |
| **LipOpt-k** | Upper bound arising from the $k$-th degree of the LP hierarchy (6) based on the sparse Krivine Positivstellenstatz. |
| **Lip-SDP** | Upper bound from Fazlyab et al. (2019) multiplied $\sqrt{d}$ where $d$ is the input dimension of the network. |
| **UBP** | Upper bound determined by the product of the layer-wise Lipschitz constants with $\ell_\infty$ metric |
| **LBS** | Lower bound obtained by sampling 50000 random points around zero, and evaluating the dual norm of the gradient |

## 7.1 EXPERIMENTS ON RANDOM NETWORKS

We compare the bounds obtained by the algorithms described above on networks with random weights and either one or two hidden layers. We define the sparsity level of a network as the maximum number of neurons any neuron in one layer is connected to in the next layer. For example, the network represented on Figure 1 has sparsity 2. The non-zero weights of network's $i$-th layer are sampled uniformly in $[-\frac{1}{\sqrt{n_i}}, \frac{1}{\sqrt{n_i}}]$ where $n_i$ is the number of neurons in layer $i$.

For different configurations of width and sparsity, we generate 10 random networks and average the obtained Lipschitz bounds. For better comparison, we plot the relative error. Since we do not know the true Lipschitz constant, we cannot compute the true relative error. Instead, we take as reference the lower bound given by **LBS**. Figures 3 and 5 show the relative error, i.e., $(\hat{L} - L_{LBS})/L_{LBS}$ where $L_{LBS}$ is the lower bound computed by **LBS** and $\hat{L}$ is the estimated upper bound. Figures 9 and 10 in Appendix C we show the values of the computed Lipschitz bounds for 1 and 2 hidden layers respectively.

When the chosen degree for **LiPopt-k** is the smallest as possible, i.e., equal to the depth of the network, we observe that the method is already competitive with the **SDP** method, especially in the case of 2 hidden layers. When we increment the degree by 1, **LiPopt-k** becomes uniformly better than **SDP** over all tested configurations. We remark that the upper bounds given by **UBP** are too large to be shown in the plots. Similarly, for the 1-hidden layer networks, the bounds from **LipSDP** are too large to be plotted.

Finally, we measured the computation time of the different methods on each tested network (Figures 4 and 6). We observe that the computation time for **LiPopt-k** heavily depends on the network sparsity, which reflects the fact that such structure is exploited in the algorithm. In contrast, the time required for **SDP** does not depend on the sparsity, but only on the size of the network. Therefore as the network size grows (with fixed sparsity level), **LipOpt-k** obtains a better upper bound and runs faster. Also, with our method, we see that it is possible to increase the computation power in order to compute tighter bounds when required, making it more flexible than **SDP** in terms of computation/accuracy tradeoff. **LiPopt** uses the Gurobi LP solver, while **SDP** uses Mosek. All methods run on a single machine with Core i7 2.8Ghz quad-core processor and 16Gb of RAM.

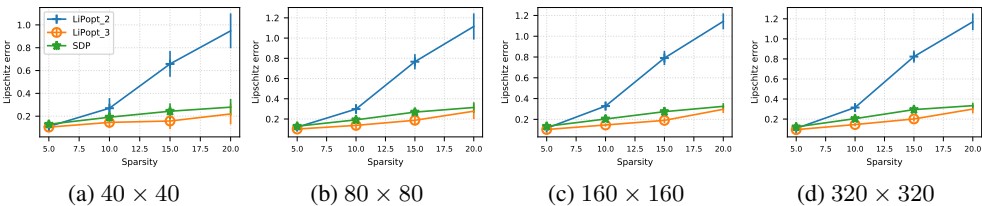

(a) $40 \times 40$   (b) $80 \times 80$   (c) $160 \times 160$   (d) $320 \times 320$

Figure 3: Lipschitz approximated relative error for 1-hidden layer networks

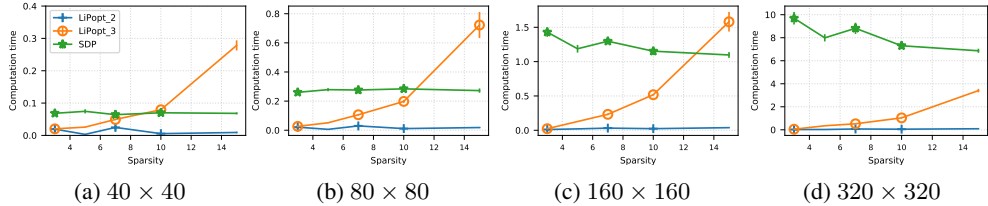

Figure 4: Computation times for 1-hidden layer networks (seconds)

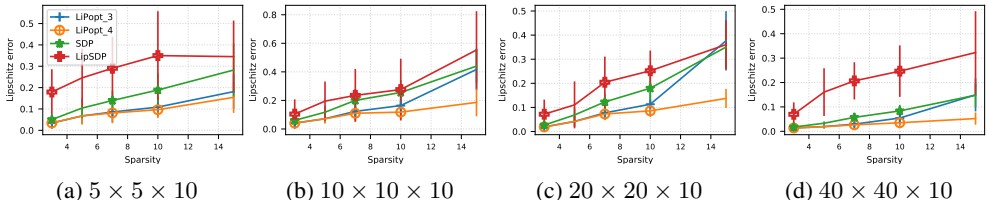

Figure 5: Lipschitz approximated relative error for 2-hidden layer networks

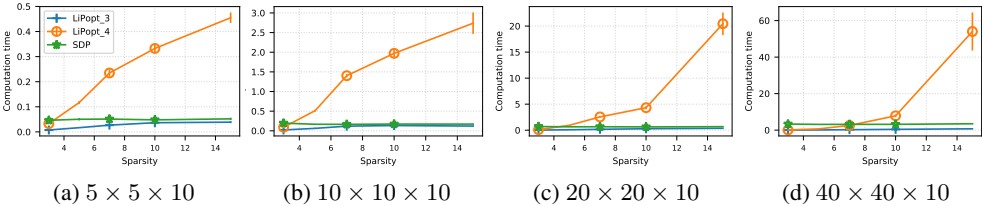

Figure 6: Computation times for 2-hidden layer networks (seconds)

## 7.2 EXPERIMENTS ON TRAINED NETWORKS

Similarly, we compare these methods on networks trained on MNIST. The architecture we use is a fully connected network with two hidden layers with 300 and 100 neurons respectively, and with one-hot output of size 10. Since the output is multi-dimensional, we restrict the network to a single output, and estimate the Lipschitz constant with respect to label 8.

Moreover, in order to improve the scalability of our method, we train the network using the pruning strategy described in Han et al. (2015)[2]. After training the full network using a standard technique, the weights of smallest magnitude are set to zero. Then, the network is trained for additional iterations, only updating the nonzero parameters. Doing so, we were able to remove 95% of the weights, while preserving the same test accuracy. We recorded the Lipschitz bounds for various methods in Table 7.2. We observe clear improvement of the Lipschitz bound obtained from **LiPopt-k** compared to **SDP** method, even when using $k = 3$. Also note that the input dimension is too large for the method **Lip-SDP** to provide competitive bound, so we do not provide the obtained bound for this method.

| Algorithm | LBS | LiPopt-4 | LiPopt-3 | SDP | UBP |
|---|---|---|---|---|---|
| Lipschitz bound | 84.2 | 88.3 | 94.6 | 98.8 | 691.5 |

## 7.3 ESTIMATING LOCAL LIPSCHITZ CONSTANTS WITH LIPOPT

In the of section 7.1, we study the improvement on the upper bound obtained by **LiPopt**, when we incorporate tighter upper and lower bounds on the variables $s_i$ of the polynomial optimization problem (4). Such bounds arise from the limited range that the pre-activation values of the network can take, when the input is limited to an $\ell_\infty$-norm ball of radius $\epsilon$ centered at an arbitrary point $x_0$.

---

[2]For training we used the code from this reference. It is publicly available in https://github.com/mightydeveloper/Deep-Compression-PyTorch

The algorithm that computes upper and lower bounds on the pre-activation values is fast (it has the same complexity as a forward pass) and is described, for example, in Wong & Kolter (2018). The variables $s_i$ correspond to the value of the derivative of the activation function. For activations like ELU or ReLU, their derivative is monotonically increasing, so we need only evaluate it at the upper and lower bounds of the pre-activation values to obtain corresponding bounds for the variables $s_i$.

We plot the local upper bounds obtained by **LiPopt-3** for increasing values of the radius $\epsilon$, the bound for the global constant (given by **LiPopt-3**) and the lower bound on the local Lipschitz constant obtained by sampling in the $\epsilon$-neighborhood (LBS). We sample 15 random networks and plot the average values obtained. We observe clear gap between both estimates, which shows that larger certified balls could be obtained with such method in the robustness certification applications.

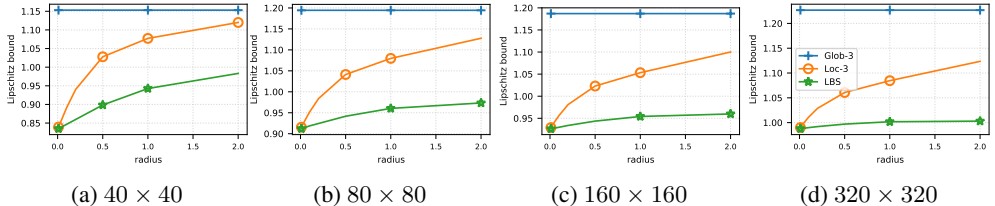

(a) $40 \times 40$    (b) $80 \times 80$    (c) $160 \times 160$    (d) $320 \times 320$

Figure 7: Global vs local Lipschitz constant bounds for 1-hidden layer networks

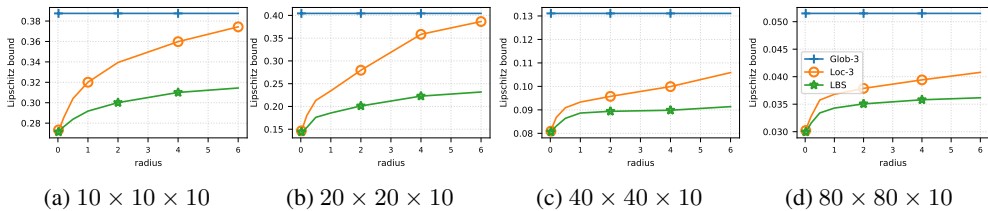

(a) $10 \times 10 \times 10$    (b) $20 \times 20 \times 10$    (c) $40 \times 40 \times 10$    (d) $80 \times 80 \times 10$

Figure 8: Global vs local Lipschitz constant bounds for 2-hidden layer networks

## 8 CONCLUSION AND FUTURE WORK

In this work, we have introduced a general approach for computing an upper bound on the Lipschitz constant of neural networks. This approach is based on polynomial positivity certificates and generalizes some existing methods available in the literature. We have empirically demonstrated that it can tightly upper bound such constant. The resulting optimization problems are computationally expensive but the sparsity of the network can reduce this burden.

In order to further scale such methods to larger and deeper networks, we are interested in several possible directions: $(i)$ divide-and-conquer approaches splitting the computation on sub-networks in the same spirit of Fazlyab et al. (2019), $(ii)$ exploiting parallel optimization algorithms leveraging the structure of the polynomials, $(iii)$ custom optimization algorithms with low-memory costs such as Frank-wolfe-type methods for SDP (Yurtsever et al., 2019) as well as stochastic handling of constraints (Fercoq et al., 2019) and $(iv)$, exploiting the symmetries in the polynomial that arise from weight sharing in typical network architectures to further reduce the size of the problems.

ACKNOWLEDGMENTS

This project has received funding from the European Research Council (ERC) under the European Union's Horizon 2020 research and innovation programme (grant agreement 725594 - time-data) and from the Swiss National Science Foundation (SNSF) under grant number 200021_178865. FL is supported through a PhD fellowship of the Swiss Data Science Center, a joint venture between EPFL and ETH Zurich. VC acknowledges the 2019 Google Faculty Research Award.

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

## A    PROOF OF THEOREM 1

**Theorem.** *Let $f$ be a differentiable and Lipschitz continuous function on an open, convex subset $\mathcal{X}$ of a euclidean space. Let $\| \cdot \|$ be the dual norm. The Lipschitz constant of $f$ is given by*

$$L(f) = \sup_{x \in \mathcal{X}} \|\nabla f(x)\|_* \tag{12}$$

*Proof.* First we show that $L(f) \leq \sup_{x \in \mathcal{X}} \|\nabla f(x)\|_*$.

$$
\begin{aligned}
|f(y) - f(x)| &= \left| \int_0^1 \nabla f((1-t)x + ty)^T (y-x)\, dt \right| \\
&\leq \int_0^1 \left| \nabla f((1-t)x + ty)^T (y-x) \right| dt \\
&\leq \int_0^1 \|\nabla f((1-t)x + ty)\|_*\, dt\, \|y - x\| \\
&\leq \sup_{x \in \mathcal{X}} \|\nabla f(x)\|_* \|y - x\|
\end{aligned}
$$

were we have used the convexity of $\mathcal{X}$.

Now we show the reverse inequality $L(f) \geq \sup_{x \in \mathcal{X}} \|\nabla f(x)\|_*$. To this end, we show that for any positive $\epsilon$, we have that $L(f) \geq \sup_{x \in \mathcal{X}} \|\nabla f(x)\|_* - \epsilon$.

Let $z \in \mathcal{X}$ be such that $\|\nabla f(z)\|_* \geq \sup_{x \in \mathcal{X}} \|\nabla f(x)\|_* - \epsilon$. Because $\mathcal{X}$ is open, there exists a sequence $\{h_k\}_{k=1}^\infty$ with the following properties:

1. $\langle h_k, \nabla f(z) \rangle = \|h_k\| \|\nabla f(z)\|_*$

2. $z + h_k \in \mathcal{X}$

3. $\lim_{k \to \infty} h_k = 0$.

By definition of the gradient, there exists a function $\delta$ such that $\lim_{h \to 0} \delta(h) = 0$ and the following holds:

$$f(z + h) = f(z) + \langle h, \nabla f(z) \rangle + \delta(h)\|h\|$$

For our previously defined iterates $h_k$ we then have

$$\Rightarrow |f(z + h_k) - f(z)| = |\|h_k\| \|\nabla f(z)\|_* + \delta(h_k)\|h_k\||$$

Dividing both sides by $\|h_k\|$ and using the definition of $L(f)$ we finally get

$$
\begin{aligned}
&\Rightarrow L(f) \geq \left| \frac{f(z + h_k) - f(z)}{\|h_k\|} \right| = |\|\nabla f(z)\|_* + \delta(h_k)| \\
&\Rightarrow L(f) \geq \lim_{k \to \infty} |\|f(z)\|_* + \delta(h_k)| = \|\nabla f(z)\|_* \\
&\Rightarrow L(f) \geq \sup_{x \in \mathcal{X}} \|\nabla f(x)\|_* - \epsilon
\end{aligned}
$$

$\square$

## B  PROOF OF PROPOSITION 1

**Proposition.** *Let $f_d$ be a dense network (all weights are nonzero). The following sets, indexed by $i = 1, \ldots, n_d$, form a valid sparsity pattern for the norm-gradient polynomial of the network $f_d$:*

$$I_i := \left\{ s_{(d-1,i)} \right\} \cup \left\{ s_{(j,k)} : \text{ there exists a directed path from } s_{(j,k)} \text{ to } s_{(d-1,i)} \text{ in } G_d \right\} \tag{13}$$

*Proof.* First we show that $\cup_{i=1}^m I_i = I$. This comes from the fact that any neuron in the network is connected to at least one neuron in the last layer. Otherwise such neuron could be removed from the network altogether.

Now we show the second property of a valid sparsity pattern. Note that the norm-gradient polynomial is composed of monomials corresponding to the product of variables in a path from input to a final neuron. This imples that if we let $p_i$ be the sum of all the terms that involve the neuron $s_{(d-1,i)}$ we have that $p = \sum_i p_i$, and $p_i$ only depends on the variables in $I_i$.

We now show the last property of the valid sparsity pattern. This is the only part where we use that the network is dense. For any network architecture the first two conditions hold. We will use the fact that the maximal cliques of a chordal graph form a valid sparsity pattern (see for example Lasserre (2006)).

Because the network is dense, we see that the clique $I_i$ is composed of the neuron in the last layer $s_{(d-1,i)}$ and all neurons in the previous layers. Now consider the extension of the computational graph $\hat{G}_d = (V, \hat{E})$ where

$$\hat{E} = E \cup \left\{ (s_{j,k}, s_{l,m}) : j, l \leq d - 2) \right\}$$

which consists of adding all the edges between the neurons that are not in the last layer. We show that this graph is chordal. Let $(a_1, \ldots, a_r, a_1)$ be a cycle of length at least 4 ($r \geq 4$). notice that because neurons in the last layer are not connected between them in $\hat{G}$, no two consecutive neurons in this cycle belong to the last layer. This implies that in the subsequence $(a_1, a_2, a_3, a_4, a_5)$ at most three belong to the last layer. A simple analysis of all cases implies that it contains at least two nonconsecutive neurons not in the last layer. Neurons not in the last layer are always connected in $\hat{G}$. This constitutes a chord. This shows that $\hat{G}_d$ is a chordal graph. Its maximal cliques correspond exactly to the sets in proposition.

$\square$

# C EXPERIMENTS ON RANDOM NETWORKS

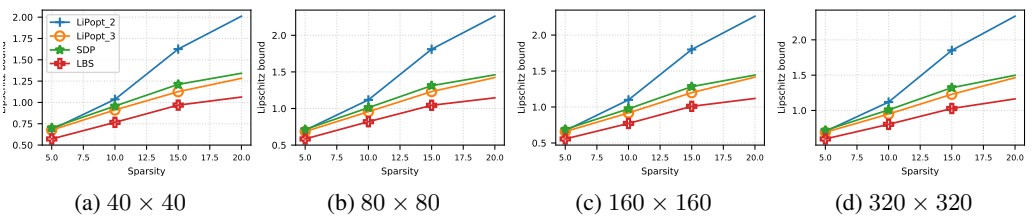

Figure 9: Lipschitz bound comparison for 1-hidden layer networks

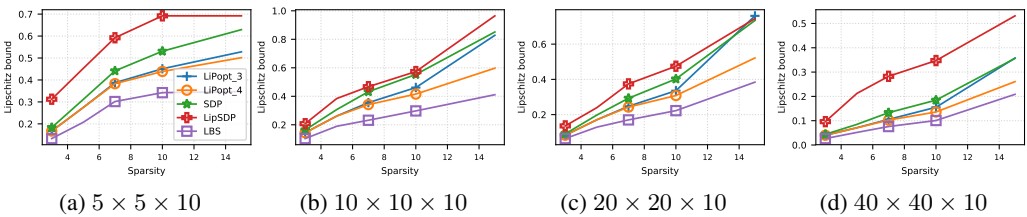

Figure 10: Lipschitz bound comparison for 2-hidden layer networks

