# OpenReview forum: "Lipschitz constant estimation of Neural Networks via sparse polynomial optimization"
_ICLR.cc/2020/Conference — Accept (Poster)_

### Official Review · AnonReviewer1 · 2019-10-20
**Official Blind Review #1**

**Rating:** 6

**Review:**

The authors study the problem of estimating the Lipschitz constant of a deep neural network with ELO activation function. The authors formulate the problem as a polynomial optimisation problem, which is elegant. Subsequently, they utilise an LP hierarchy based on the Krivine-Vasilescu-Handelman’s Positivstellensatz and suggest exploiting sparsity therein. The computational results are clearly not sufficient to apply this approach to real-world neural networks, but are still respectable.

Section 3 (Theorem 2) is not original work, as leaving the theorem without a reference would imply: the authors cite Section 9 of Lasserre's 2015 book later, so they are clearly aware of this, and there are many application even within verification, e.g.,
https://link.springer.com/content/pdf/10.1007%2F978-3-319-48989-6_44.pdf
https://ieeexplore.ieee.org/document/8493559

The suggestions as to the exploitation of sparsity (Section 4) are not original work either. The authors could cite, e.g., JB Lasserre: Convergent SDP-relaxations in polynomial optimization with sparsity (SIAM Journal on Optimization, 2006), as one of the early proponents of the exploitation of sparsity.

In Section 7:
-- The claim "We observed clear improvement of the Lipschitz bound obtained, compared to the SDP method" is not supported by the results the authors present.
-- The authors do not present the run-time. This needs to be included, considering they imply that the key improvement over the traditional SDP is that this works with smaller variables and should be faster.
-- The presentation of the experimental results should be improved, so as to follow the NIPS reproducibility checklist, or at least have error bars at one standard deviation and standard deviation in the table.

Other than that, the paper is well written (modulo Section missing in "Section 5" at the top of Section 7), and I would recommend its acceptance.

**Experience Assessment:**

I have published in this field for several years.

**Review Assessment: Checking Correctness Of Derivations And Theory:**

I carefully checked the derivations and theory.

**Review Assessment: Checking Correctness Of Experiments:**

I carefully checked the experiments.

**Review Assessment: Thoroughness In Paper Reading:**

I read the paper thoroughly.

---

> ### Author Response · Authors · 2019-11-14
> **Updated version with new experiments**
>
> Dear Reviewer1, we have uploaded a revised version of the paper addressing your concerns:
>
> 1. In section 7.1 we have added computation time plots. We see the time is lower for LipOpt when sparsity is increased. LiPopt with higher degree and less sparsity can take more time than SDP BUT it obtains bounds which are more tight. i.e.
> if one is ok with less tight bounds, LiPopt with low degree is faster than SDP, but if time is available LiPopt allows to trade more computation for accuracy.
>
> We also remark that before our work, the SDP method was limited to one-hidden-layer networks. As part of our contribution we generalize this method to arbitrary number of layers
>
> 2. Also in section 7.1, have added error bars (time and Lipschitz constant estimation).
>
> We hope this will reassure you about the quality of the work and that you consider raising your score. We thank you again for your contribution to the improvement of the paper.

---

### Official Review · AnonReviewer3 · 2019-10-23
**Official Blind Review #3**

**Rating:** 8

**Review:**

This paper presents a general approach for upper bounding the Lipschitz constant of a neural network by relaxing the problem to a polynomial optimization problem. And the authors extend the method to fully make use of the sparse connections in the network so that the problem can be decomposed into a series of much smaller problems, saving large amount of computations and memory. Even for networks that don't have high-level sparse connections, the proposed method can still help to reduce the size of the problem. This paper also compares the proposed LiPopt method with another solution derived from a quadratically constrained quadratic program reformulation. Compared with this method, the LiPopt method can handle cases with more parameters efficiently.

Calculating a TIGHT upper bound of a neural network efficiently is very valuable and useful in many areas in deep learning community. And I really like the potential to use this LiPopt method to upper bound local Lipschitz constant in a given neighboring region, which will be very useful in certificated robustness application, etc..

I also like that the authors present results on networks trained on real-world dataset (MNIST). My only suggestion is that I'd like to see LiPopt's computation time and memory usage compared to its counterparts, as the authors argue the proposed method can fully exploit the sparse connections to reduce the problem size.

=======
Update: I am satisfied with the authors' solid response and would like to raise my score.

**Experience Assessment:**

I have read many papers in this area.

**Review Assessment: Checking Correctness Of Derivations And Theory:**

I assessed the sensibility of the derivations and theory.

**Review Assessment: Checking Correctness Of Experiments:**

I assessed the sensibility of the experiments.

**Review Assessment: Thoroughness In Paper Reading:**

I read the paper at least twice and used my best judgement in assessing the paper.

---

> ### Author Response · Authors · 2019-11-14
> **Updated version with new experiments**
>
> Dear Reviewer3, we have uploaded a revised version of the paper addressing your concerns:
>
> 1. We have included a new section (7.3) comparing the Local Lipschitz constant bounds that we can obtain, with the
> global constant, we find that they can differ quite a bit. Hence, they potentially provide larger certified regions around samples, in the context of certified robustness of networks.
>
> 2. In section 7.1 we have added computation time plots. We see the time is lower for LipOpt when sparsity is increased. LiPopt with higher degree and less sparsity can take more time than SDP BUT it obtains bounds which are more tight. i.e.
> if one is ok with less tight bounds, LiPopt with low degree is faster than SDP, but if time is available LiPopt allows to trade more computation for accuracy.
>
> We also remark that before our work, the SDP method was limited to one-hidden-layer networks. As part of our contribution we generalize this method to arbitrary number of layers.
>
>
> We hope this will reassure you about the quality of the work and that you consider raising your score. We thank you again for your contribution to the improvement of the paper.

---

### Official Review · AnonReviewer2 · 2019-10-24
**Official Blind Review #2**

**Rating:** 6

**Review:**

In this paper, the authors introduce a framework for computing upper bounds on the Lipschitz constant for neural nets. The main contribution of the paper is to leverage the sparsity properties of typical feed-forward neural nets to reduce the computational complexity of the algorithm. Through experiments, the authors show that the proposed algorithm computes tighter Lipschitz bounds compared to baselines.

The approach proposed in the paper looks interesting. Although the presentation can be made clearer in places. For example, in equation (4), it would be helpful to explicitly state over which parameters the max is taken. There's also a number of small typos that need to be fixed. For example: "We refer to d as the depth, and we we focus on the case where fd has a single real value as output." on page 1.

I found the proposed algorithm and the discussions in Section 2 and 3 interesting, although I am not familiar enough with the literature on polynomial optimization to evaluate whether there is any significantly new idea presented in these sections. I found section 4 very interesting too, and very important towards making the algorithm actually computationally tractable. I have a couple of concerns with the rest of the paper however, which `I describe below:

1. It is nice that upper bounds for the local Lipschitz constant can be incorporated easily into the formulation. I would have liked to see some experiments on evaluating local Lipschitz constants though, and how they compare with other methods, since this is a very popular setting in which such techniques are used nowadays.

2. The paper overall I think would benefit from a better experimental evaluation. It would be interesting to see how much the sparsity pattern in convnets affect results compared to other baselines. It would also be interesting to see how the bound degrades as the network grows bigger, and in particular as the depth increases.

Given the lack of thorough experiments in the paper, I am giving the paper a borderline rating. I am however willing to increase my score based on discussions with the authors and other reviewers.

===================================

Edit after rebuttal:
The latest draft addresses most of my concerns, and I am happy to recommend accepting this paper now.

**Experience Assessment:**

I do not know much about this area.

**Review Assessment: Checking Correctness Of Derivations And Theory:**

I assessed the sensibility of the derivations and theory.

**Review Assessment: Checking Correctness Of Experiments:**

I carefully checked the experiments.

**Review Assessment: Thoroughness In Paper Reading:**

I read the paper at least twice and used my best judgement in assessing the paper.

---

> ### Author Response · Authors · 2019-11-14
> **Updated version with new experiments**
>
> Dear Reviewer2, we have uploaded a revised version of the paper addressing your concerns:
>
> 1. In equation (4), the parameters over which the max is taken are in the description of the set, they are the variables 0<= s_i <= 1, and -1 <= t_i <=1.
>
> 2. We have included a new section (7.3) comparing the Local Lipschitz constant bounds that we can obtain, with the
> global constant, we find that they can differ quite a bit. Hence, they potentially provide larger certified regions around samples, in the context of certified robustness of networks.
>
> We will consider adding an experiment on how the bounds change as the depth increases for the final version of the paper.
>
> We hope this will reassure you about the quality of the work and that you consider raising your score. We thank you again for your contribution to the improvement of the paper.

---

### Author Response · Authors · 2019-11-07
**Author response to Reviewers**

We thank the reviewers for their feedback, and we address their concerns:

First, we have uploaded a new version of the paper correcting the following:

1. Fixed some typos pointed out by Reviewer2 and Reviewer1

2. As was asked by Reviewer1, we have added references in the statement of Theorem 2, which is a classical result in algebraic geometry and we only include for completeness (we do not claim it is our original work) as well as
Theorem 3, as our work leverages such results to implement the algorithm for upper bounding the Lipschitz constant. We remark that they are adapted to our particular setting, as the fully general result is not needed and we think it might hurt the readability and accessibility of the paper to a broad audience.

We have made this more explicit in the new version and we hope that Reviewer1 agrees it is better understood now that Thm2 and Thm3 are not presented as original work.

3. At the end of section 4, we added more references to the use of sparsity in polynomial optimization (which we don't claim to be part of our contribution), as suggested by Reviewer1. Our work shows that the sparsity of the neural network is directly linked to the sparsity of its norm-gradient polynomial, which in turn allows the use of the sparse-polynomial optimization methods. We have made this more clear in the last paragraph in section 4.

4. We included references to other applications of sparse polynomial optimization in safety verification, as suggested by Reviewer1. We are thankful for improving the completeness of our bibliography.

5. We added the bound obtained by LiPopt in our MNIST example, with relaxation degree 4. The bound is tighter.

Secondly, we will add the following as soon as possible:

1.  We will plot how the estimate of Local Lipschitz constant improves over the global Lipschitz constant, when we evaluate over an l_infinity ball centered at some particular point x_0, with varying radius epsilon. In this way we aim to address the interest of Reviewer2 and Reviewer3 in this application of our proposed method.

2. We have found that the naive upper bound degrades considerably when the depth is increased, while the upper bounds we compute degrade at a much slower rate. So overall, the improvement over the naive upper bound that we can obtain with our method becomes more pronounced with increased depth. We will include an experiment and discussion on this phenomenon. In this way we aim to address a concern of Reviewer2.

3. We will add error bars on our plots. However, this needs a small change in the plots as we now explain. When randomizing over the neural network's weights, the Lipschitz constant naturally becomes a random variable, and the  error bars would show this. In order to provide meaningful error bars we want to plot the variability of the approximation error, but this would require access to the true lipschitz constant, which we can not obtain. We will use the lower bound obtained by sampling as an estimate of the error. This should provide a better sense of how much of an improvement does the LP method has over the SDP. In this way we will address the concern of Reviewer1 about the NeurIPS reproducibility checklist.

4. We will compare average solving time for LiPopt with relaxation degree 2, 3 and 4, for one and two hidden layers, compared to the SDP approach. At lower degrees of relaxation we have observed it is faster to solve an LP rather than an SDP and it can obtain better bounds. If we consider higher degrees of relaxation we observe the LP can take more time to solve BUT it obtains increasingly tighter bounds on the Lipschitz constant, thus providing a way to trade off more computation time with accuracy. We will also asses the memory consumption.

With this we will hopefully improve our paper and make our claims and contributions stronger, as suggested by the reviewers.

A final remark:
The SDP method, although is not the one we suggest to use due to the scalability of commercial SDP solvers, we consider also part of our algorithmic contribution. Although first proposed by Raghunathan et al. 2018a, In its original form it was limited to the one hidden layer case. In our work we show how it extends also to the multilayer case. Note that the limitation to the one hidden layer case is pointed out as a main drawback by subsequent work (Raghunathan et al. 2018b) among other works, so we think that lifting this drawback is a valuable contribution.
We make this point in section 5 (page 7).

References:

Aditi Raghunathan, Jacob Steinhardt, and Percy Liang. Certified defenses
against adversarial examples. In International Conference on Learning
Representations, 2018a. URL https: //openreview.net/forum?id=Bys4ob-Rb.

Aditi Raghunathan, Jacob Steinhardt, and Percy S Liang. Semidefinite
relaxations for certifying robustness to adversarial examples. In Advances in
Neural Information Processing Systems, pp. 10877–10887, 2018b.

---

> ### Comment · AnonReviewer1 · 2019-11-08
> **Thank you**
>
> Many thanks for the changes to the draft, which have improved it substantially.
>
> I am quite happy with the response as well. I imagine that you could perhaps just vary the seed to produce the error bars:
>     seed = 7
>     np.random.seed(seed)
>     torch.manual_seed(seed)
> rather than necessarily vary the neural network otherwise. This would already account for the randomisation in the stochastic gradient descent or similar.

---

### Decision · Program_Chairs · 2019-12-19

**Decision:**

Accept (Poster)

**Comment:**

This paper improves upper bound estimates on Lipschitz constants for neural networks by converting the problem into a polynomial optimization problem.  The proposed method also exploits sparse connections in the network to decompose the original large optimization problem into smaller ones that are more computationally tractable. The bounds achieved by the method improve upon those found from a quadratic program formulation.  The method is tested on networks with random weights and networks trained on MNIST and provides better estimates than the baselines.

The reviews and the author discussion covered several topics.  The reviewers found the paper to be well written.  The reviewers liked that tighter bounds on the Lipschitz constants can be found in a computationally efficient manner.  They also liked that the method was applied to a real-world dataset, though they noted that the sizes of the networks analyzed here are smaller than the ones in common use.  The reviewers pointed out several ways that the paper could be improved.  The authors adopted these suggestions including additional comparisons, computation time plots, error bars, and relevant references to related work.  The reviewers found the discussion and revised paper addressed most of their concerns.

This paper improves on existing methods for analyzing neural network architectures and it should be accepted.